# Genetic basis of falling risk susceptibility in the UK Biobank Study

Katerina Trajanoska [1], Lotta J. Seppala[2], Carolina Medina-Gomez [1], Yi-Hsiang Hsu[3,4,5], Sirui Zhou[6], Natasja M. van Schoor[7], Lisette C. P. G. M. de Groot [8], David Karasik [3,9], J. Brent Richards [6,10,11], Douglas P. Kiel [3,4,12], Andre G. Uitterlinden [1,2], John R. B. Perry[1,13], Nathalie van der Velde[2], Felix R. Day [1,13,14] & Fernando Rivadeneira [1,14✉]

Both extrinsic and intrinsic factors predispose older people to fall. We performed a genome-wide association analysis to investigate how much of an individual's fall susceptibility can be attributed to genetics in 89,076 cases and 362,103 controls from the UK Biobank Study. The analysis revealed a small, but significant SNP-based heritability (2.7%) and identified three novel fall-associated loci ($P_{combined} \leq 5 \times 10^{-8}$). Polygenic risk scores in two independent settings showed patterns of polygenic inheritance. Risk of falling had positive genetic correlations with fractures, identifying for the first time a pathway independent of bone mineral density. There were also positive genetic correlations with insomnia, neuroticism, depressive symptoms, and different medications. Negative genetic correlations were identified with muscle strength, intelligence and subjective well-being. Brain, and in particular cerebellum tissue, showed the highest gene expression enrichment for fall-associated variants. Overall, despite the highly heterogenic nature underlying fall risk, a proportion of the susceptibility can be attributed to genetics.

[1] Department of Internal Medicine, Erasmus MC University Medical Center, Rotterdam, The Netherlands. [2] Department of Internal Medicine, Section of Geriatric Medicine, Academic Medical Center, University of Amsterdam, Amsterdam, The Netherlands. [3] Hinda and Arthur Marcus Institute for Aging Research, Hebrew SeniorLife, Boston, MA, USA. [4] Broad Institute of MIT and Harvard, Boston, MA, USA. [5] Molecular and Integrative Physiological Sciences, Harvard School of Public Health, Boston, MA, USA. [6] Lady Davis Institute, Jewish General Hospital, McGill University, Montréal, Québec, Canada. [7] Department of Epidemiology and Biostatistics, Amsterdam Public Health Research Institute, VU University Medical Center, Amsterdam, The Netherlands. [8] Wageningen University, Division of Human Nutrition, PO-box 17, 6700 AA Wageningen, The Netherlands. [9] Azrieli Faculty of Medicine, Bar-Ilan University, Safed, Israel. [10] Department of Human Genetics, McGill University, Montréal, Québec, Canada. [11] Departments of Medicine and Epidemiology, Biostatistics and Occupational Health, McGill University, Montréal, Québec, Canada. [12] Department of Medicine, Beth Israel Deaconess Medical Center and Harvard Medical School, Boston, MA, USA. [13] MRC Epidemiology Unit, University of Cambridge School of Clinical Medicine, Cambridge, UK. [14] These authors contributed equally: Felix R. Day, Fernando Rivadeneira. ✉email: f.rivadeneira@erasmusmc.nl

F alls are a growing healthcare problem in older adults. They are a major contributor to immobility and premature nursing home placement[1]. Furthermore, they are a leading cause of unintentional injuries, which require medical treatment[2], and increase the demand on healthcare resources. At present, between 0.85% and 1.50% of the total healthcare expenditures in Europe, North America, and Australia are fall-related costs[3]. As the global population continues to grow and become older, healthcare costs related to falls will grow accordingly[4].

There are numerous extrinsic and intrinsic factors predisposing older adults to fall, which have been intensively studied in the past decades[5–7]. A number of the intrinsic ones, in particular postural balance, gait speed, muscle function, and cognition, have a recognized heritable component[8–10], suggesting that investigation into the genetic influence on falls may be warranted. Pharmacogenetic variability may also contribute to drug-induced falls as a result on the variability of drug responses and risk of adverse effects of medications[11,12]. Twin studies have found that familial factors, consisting of genetic and shared environmental influences, explain about 35% of the variability in the likelihood of experiencing at least one incident fall and 45% of the variability in the risk for recurrent falls[13]. Genetic variation is stable across the human lifespan and identification of genetic factors for falling may help optimizing effective fall prevention programs, i.e., improve fall risk stratification, while also providing biological insight into their etiology. So far, few studies have been performed to identify genetic factors underlying fall risk, likely due to lack of a well-powered discovery setting. In a candidate-gene study without replication ($N_{cases}$ = 955; $N_{total}$ = 4163), Judson et al.[14] reported that female carriers of the ACTN3 genetic variant (rs1815739), which is associated with reduced muscle mass and force, had 33% higher risk of falling compared to non-carriers. However, to date, no genome-wide association studies (GWASs) have been performed to identify genetic variants associated with increased fall risk in a hypothesis-free context.

Here we undertook the first large-scale GWAS on falling risk in 89,076 cases and 362,103 controls from the UK Biobank Study. We identified three novel fall-associated loci. Our follow-up analyses showed falling risk to be a heritable (2.7%), polygenic, and highly heterogeneous trait genetically correlated with fracture risk and grip strength among other traits. Brain and, in particular, the cerebellum tissue showed the highest gene expression enrichment (false discovery rate < 5%) for fall-associated variants. Overall, our study provides novel insights into the genetic landscape of falling risk susceptibility.

## Results

**GWAS study of falling.** Our study included data from 451,179 (89,076 cases) White European individuals (40–69 years) from the UK Biobank. Fall cases were defined as participants who gave positive answer to the following question "In the last year have you had any falls?" We tested 7,745,390 million variants (minor allele frequency (MAF) > 0.01, imputation quality > 0.3) for association with fall risk. We identified two loci associated with fall risk mapping to 7p21.3 near PER4 (rs2709062-A, odds ratio (OR) = 1.03, $P = 3.4 \times 10^{-8}$) (Fig. 1a) and 19q12 near TSHZ3 (rs2111530-G, OR = 1.03, $P = 1.2 \times 10^{-8}$) (Fig. 1b). Moreover, 58 single-nucleotide polymorphisms (SNPs) were associated at $P < 5 \times 10^{-6}$ of which 15 were associated at genome-wide suggestive level ($P < 5.0 \times 10^{-7}$) (Table 1 and Supplementary Fig. 1). Linkage disequilibrium score regression (LDSR) showed no sign of genomic inflation (Intercept = 1.01) compatible with a polygenic architecture of the trait and no evidence for stratification (Supplementary Fig. 2). Using a genome-wide gene-based approach, implemented by MAGMA, we identified nine fall-associated

genes (Supplementary Fig. 3); the MAGMA gene-set results were later served as an input for the tissue expression analysis. The majority of the SNPs with $P < 5 \times 10^{-6}$ (Supplementary Fig. 4A) were located in intergenic or intronic regions and >70% of the variants overlapped chromatin state annotations (Supplementary Fig. 4B) of potential involvement in gene regulation; however, only 3.3% possessed strong (regulomeDB score ≤ 2) regulatory potential (Supplementary Fig. 4C). The 7p21.3 risk locus did not harbor genes with relevant expression quantitative trait loci (eQTL) and/or chromatin interactions (Supplementary Fig. 5A). The TSHZ3 gene at the 19q12 locus was annotated by eQTLs in the thyroid tissue and was also implicated by chromatin interactions in the mesendodermal tissue and the mesenchymal stem cells (Supplementary Fig. 5B). However, none of the lead SNPs showed any evidence for eQTL effects. In addition, several suggestive SNPs were associated with body composition measures such as body mass index (BMI), fat mass, and fat-free mass (Supplementary Table 1).

**Replication.** We took forward for replication the 17 genome-wide suggestive SNPs ($P < 5.0 \times 10^{-7}$) from the discovery UK Biobank (UKBB) sample; we sought replication in two smaller prospective population-based studies with older participants, namely the Rotterdam Study (1009 cases and 4925 controls) and B-PROOF (1206 cases and 1364 controls) cohorts. The B-PROOF Study is a clinical trial on B-vitamin supplements in older adults of advanced age (mean age 74.1 ± 6.5 years) in which fall risk was assessed using retrospective questionnaires at baseline and prospective fall calendars[15]. The Rotterdam Study is a population-based cohort with fall information from participants (mean age 69.5 ± 9.2 years) collected retrospectively from baseline questionnaires[16]. We defined replication as loci harboring variants associated at a nominal ($P < 0.05$) significant threshold in the replication setting or reaching the genome-wide significant (GWS) ($P < 5.0 \times 10^{-8}$) threshold in the combined meta-analysis. Overall, the top two SNPs from the discovery phase remained GWS significant in the combined meta-analysis, while replication also brought one additional locus (5q21.3) mapping to RP11-6N13.1 above the GWS threshold (Table 1 and Fig. 1c). The lead SNP in this locus did not show any evidence for eQTL effects (Supplementary Fig. 5c).

**Polygenic risk scores.** Next, we evaluated the ability of polygenic risk scores (PRSs) constructed from the UK Biobank GWAS results to discriminate between fallers and non-fallers in two independent prospective cohorts. We hypothesized falling risk to follow a polygenic mode inheritance, i.e., is influenced by numerous genes with small individual effects[17] and, hence, non-GWS SNPs may also contribute to the genetic component of falling risk. Therefore, PRSs were constructed using PRSice[18] for a series of P-value thresholds ranging from $5 \times 10^{-8}$ to 1. Variants ($P < 1$, MAF > 0.05, and imputation quality > 0.3) were clumped before analysis ($r^2 < 0.05$, window: 300 kb) to obtain the most significant SNP in the locus. In line with polygenic inheritance, the PRS explained a small, but robust, proportion of the trait variance (max $R^2 = 0.29\%$) along different P-value thresholds. In the B-PROOF Study, the PRS derived from GWS variants ($P \leq 5 \times 10^{-8}$) was associated with prospective falls (reported by fall calendar) and explained the largest fraction of the trait variance (max $R^2 = 0.29\%$) (Supplementary Data 1 and Fig. 2a). In contrast, the PRSs constructed from variants in the lower significance thresholds ($P \leq 0.03$) were the ones more strongly associated with retrospectively collected falls (reported by baseline questionnaires) (Supplementary Data 1 and Fig. 2b, c).

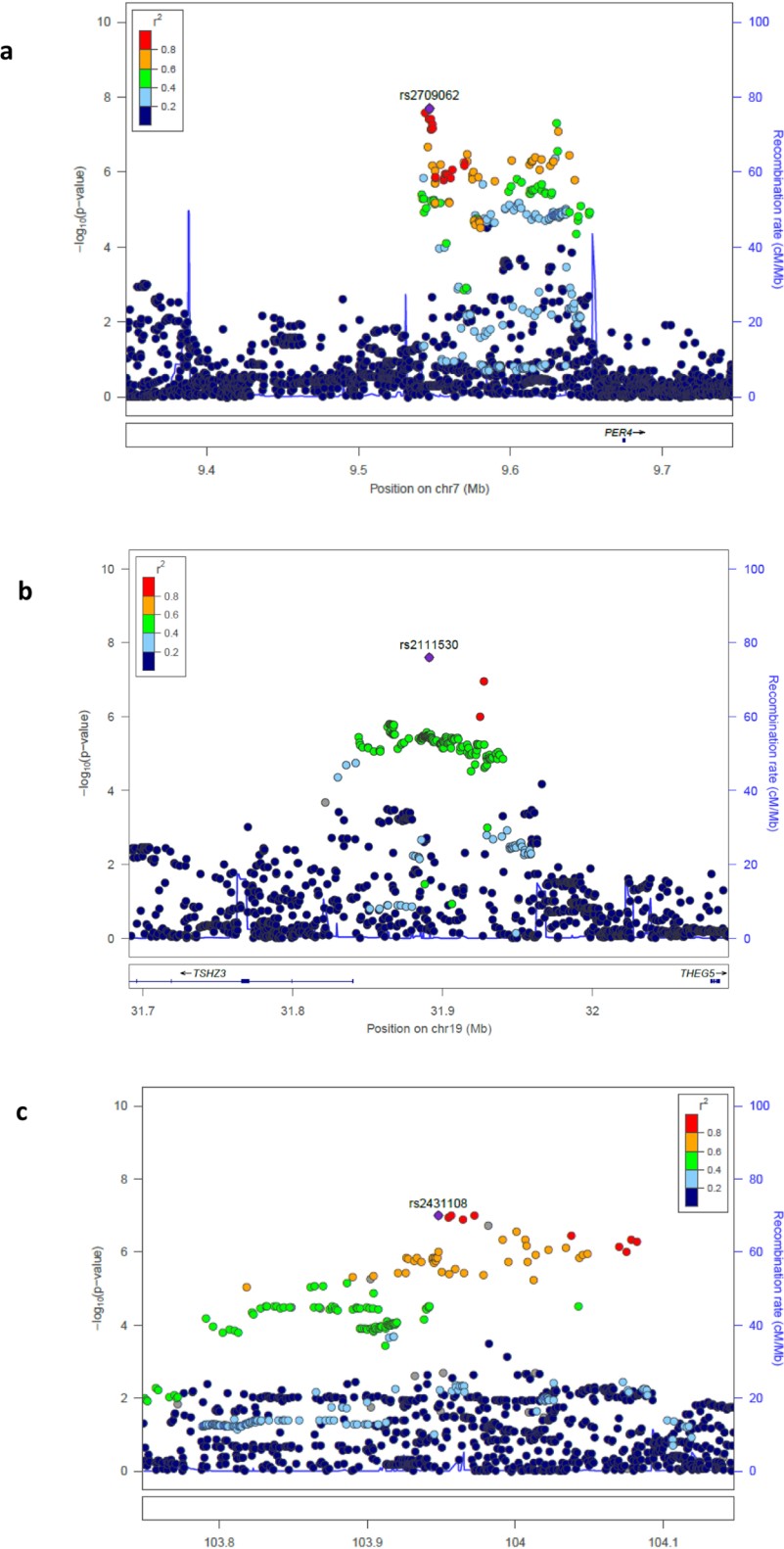

**Fig. 1 Regional association plots of the associated loci.** In each plot, the −log₁₀ of *p*-values are on the left *y*-axis, the SNP genomic position (hg19) on the *x*-axis, and the estimated recombination rate from 1000 genomes (March 2015 EUR) are on the right *y*-axis and plotted in blue. SNPs are colored red to reflect linkage disequilibrium (LD) with the most significant SNP in purple (pairwise $r^2$ from 1000 genomes March 2014 EUR). (**a**) 7p21.3, (**b**) 19q12, and (**c**) 5q21.2 loci.

**Table 1 Lead SNPs of loci associated with increased risk of falling ($p < 5 \times 10^{-7}$).**

| Locus | Annotation | Closest gene | Position | SNP | EA | NEA | EAF | UKBB N = 89,076/362,103 OR (95% CI) | p | Rotterdam Study N = 1009/4925 OR (95% CI) | p | B-PROOF N = 1206/1364 OR (95% CI) | p | Combined N = 91,219/368,392 OR (95% CI) | p |
|---|---|---|---|---|---|---|---|---|---|---|---|---|---|---|---|
| Genome-wide significant loci ($P \leq 5 \times 10^{-8}$) | | | | | | | | | | | | | | | |
| 5q21.2 | | RP11-6N13.1 | 103947968 | rs2431108 | C | T | 0.33 | 1.03 (1.02-1.04) | $9.9 \times 10^{-8}$ | 1.13 (1.03-1.25) | 0.01 | 0.98 (0.88-1.10) | 0.77 | 1.03 (1.02-1.04) | $4.20 \times 10^{-8}$ |
| 7p21.3 | Intergenic | PER4 | 9546806 | rs2709062 | A | G | 0.50 | 1.03 (1.02-1.04) | $2.4 \times 10^{-8}$ | 1.08 (0.98-1.19) | 0.12 | 1.13 (1.01-1.26) | 0.03 | 1.03 (1.02-1.04) | $4.04 \times 10^{-9}$ |
| 19q12 | Intergenic | TSHZ3 | 31891006 | rs2111530 | G | A | 0.39 | 1.03 (1.02-1.04) | $2.5 \times 10^{-8}$ | 1.00 (0.91-1.11) | 0.88 | 1.08 (0.86-1.18) | 0.20 | 1.03 (1.02-1.04) | $1.82 \times 10^{-8}$ |
| Genome-wide suggestive loci ($P < 5 \times 10^{-7}$) | | | | | | | | | | | | | | | |
| 1p13.3 | Intergenic | NTNG1 | 107666942 | rs76259395 | A | G | 0.03 | 1.03 (1.02-1.05) | $6.6 \times 10^{-8}$ | 1.03 (0.90-1.19) | 0.64 | 1.02 (0.82-1.26) | 0.86 | 1.03 (1.02-1.05) | $6.46 \times 10^{-8}$ |
| 1p13.2 | Intronic | FAM212B | 112274162 | rs6658723 | T | C | 0.02 | 1.02 (1.01-1.03) | $2.6 \times 10^{-7}$ | 1.00 (0.91-1.11) | 0.96 | 1.01 (0.90-1.11) | 0.93 | 1.02 (1.01-1.03) | $3.14 \times 10^{-7}$ |
| 2p16.1 | Intergenic | EIF3FP3 | 59295476 | rs67174662 | A | C | 0.02 | 1.02 (1.01-1.03) | $5.0 \times 10^{-7}$ | 0.99 (0.90-1.10) | 0.91 | 1.02 (0.91-1.14) | 0.77 | 1.02 (1.01-1.03) | $5.56 \times 10^{-7}$ |
| 2p16.1 | Intronic | BCL11A | 60333030 | rs974135 | T | C | 0.33 | 1.02 (1.02-1.03) | $3.9 \times 10^{-7}$ | 0.96 (0.87-1.07) | 0.51 | 0.94 (0.83-1.06) | 0.34 | 1.02 (1.01-1.03) | $9.45 \times 10^{-7}$ |
| 3p14.2 | Intronic | FHIT | 60138226 | rs7616516 | A | G | 0.07 | 1.05 (1.03-1.06) | $2.3 \times 10^{-7}$ | 0.95 (0.79-1.14) | 0.56 | 0.99 (0.85-1.16) | 0.92 | 1.04 (1.03-1.06) | $3.49 \times 10^{-7}$ |
| 5q35.2 | Intergenic | DRD1 | 174888896 | rs2471020 | C | T | 0.59 | 1.02 (1.01-1.03) | $4.8 \times 10^{-7}$ | 1.07 (0.97-1.17) | 0.20 | 1.03 (0.92-1.16) | 0.61 | 1.02 (1.01-1.03) | $2.43 \times 10^{-7}$ |
| 6p21.1 | Intronic | TRERF1 | 42360455 | rs72857666 | T | C | 0.03 | 1.07 (1.04-1.10) | $1.5 \times 10^{-7}$ | 0.83 (0.61-1.14) | 0.26 | 0.96 (0.68-1.36) | 0.81 | 1.07 (1.04-1.09) | $2.85 \times 10^{-7}$ |
| 7p21.3 | Intergenic | NXPH1 | 9629549 | rs12666565 | C | T | 0.18 | 1.03 (1.02-1.04) | $4.7 \times 10^{-7}$ | 1.06 (0.93-1.21) | 0.41 | 1.13 (0.98-1.31) | 0.10 | 1.03 (1.02-1.04) | $1.77 \times 10^{-7}$ |
| 11p14.1 | Intronic | BDNF | 27643725 | rs11030084 | T | C | 0.03 | 1.03 (1.02-1.04) | $1.1 \times 10^{-7}$ | 1.10 (0.96-1.25) | 0.16 | 0.88 (0.78-0.98) | 0.02 | 1.03 (1.02-1.04) | $8.92 \times 10^{-8}$ |
| 11p14.1 | Intronic | MPPED2 | 30492581 | rs494221 | A | G | 0.03 | 1.03 (1.02-1.04) | $6.7 \times 10^{-8}$ | 0.98 (0.88-1.08) | 0.68 | 0.95 (0.83-1.07) | 0.55 | 1.02 (1.01-1.03) | $3.01 \times 10^{-7}$ |
| 11p15.5 | Intronic | TSPAN4 | 855372 | rs28672671 | C | T | 0.03 | 1.03 (1.02-1.04) | $5.1 \times 10^{-8}$ | 0.95 (0.83-1.07) | 0.27 | 0.89 (0.78-1.01) | 0.08 | 1.03 (1.02-1.04) | $2.23 \times 10^{-7}$ |
| 14.q21.2 | Intergenic | RPL10L | 46984874 | rs12848871 | C | T | 0.03 | 1.03 (1.02-1.04) | $1.4 \times 10^{-7}$ | 1.05 (0.94-1.14) | 0.37 | 0.97 (0.86-1.09) | 0.59 | 1.03 (1.02-1.04) | $1.44 \times 10^{-7}$ |
| 19q12 | Intronic | ZNF536 | 30772256 | rs28633123 | T | C | 0.21 | 1.03 (1.02-1.04) | $9.4 \times 10^{-8}$ | 1.05 (0.93-1.20) | 0.37 | 1.00 (0.87-1.16) | 0.96 | 1.03 (1.02-1.04) | $7.29 \times 10^{-7}$ |
| 20p11.23 | Intronic | CTNNBL1 | 36382855 | rs6063547 | G | T | 0.03 | 1.03 (1.02-1.04) | $6.8 \times 10^{-8}$ | 1.06 (0.92-1.21) | 0.44 | 0.99 (0.85-1.16) | 0.92 | 1.03 (1.02-1.04) | $6.45 \times 10^{-8}$ |

CI confidence interval, EA effect allele, EAF effect allele frequency, NEA non-effect allele, OR odds ratio; p p-value of the SNP falls association.
Lead SNP is defined as SNP with the lowest p-value.

**Individual and shared heritability of falls**. We then used LDSR to estimate the heritability (individual and shared) between falls and different diseases and traits[19], restricting the analysis to common variants (MAF > 5%) present in HapMap3. We considered traits that are closely phenotypically related with fall risk such as musculoskeletal, neurological, and psychological ones, and use of a variety of medications. As expected, the SNP-based heritability of falls was low ($h^2 = 0.027$, SE = 0.002). In relation to other traits (Supplementary Data 2 and Fig. 3), falling had a positive genetic correlation with fracture ($r_g = 0.45$, SE = 0.05, $p = 5.0 \times 10^{-21}$) and was negatively genetically correlated ($r_g = -0.17$, SE = 0.04, $p = 1.0 \times 10^{-5}$) with muscle strength. Of particular note, there was no evidence of any genetic correlation between fall risk with bone mineral density (BMD) or lean mass. We also used LDSR to explore whether falls are genetically correlated with a range of neurological, psychiatric, and behavioral traits (Supplementary Data 2 and Fig. 3). Falls were strongly positively correlated with insomnia, neuroticism, depressive symptoms, and attention deficit hyperactivity disorders, although the latter two did not pass Bonferroni correction (0.05/49 = 0.001). We also observed a small-to-moderate negative correlation with intelligence/IQ ($r_g = -0.12$, SE = 0.04, $p = 0.0012$) and subjective well-being ($r_g = -0.30$, SE = 0.05, $p = 6.4 \times 10^{-6}$) (Supplementary Data 2 and Fig. 3). Medication use is a well-established risk factor for falls, either directly (affecting balance, attention, or muscle tone) or indirectly (as proxies of underlying conditions influencing the risk of falling; joint pain, arthrosis, cardiovascular diseases among many others). A recent GWAS in ~320,000 individuals from the UK Biobank[20] identified 505 independent genetic loci associated with medication use grouped across 23 categories. Using this data, we observed positive genetic correlations between fall risk and use of medication such as opioids, anti-inflammatory and anti-rheumatic drugs, anilids, and drugs for peptic ulcer and gastro-esophageal reflux disease.

**Enrichment of gene expression across tissues**. Further, to quantify the enrichment of the fall-associated SNP signals across different tissues, we used an extension of LDSR, namely stratified LDSR (LDSR to specifically expressed genes, LDSC-SEGs), and identified significant enrichment confined to tissues from the central nervous system and particularly those derived from the cerebellum (Supplementary Data 3 and Fig. 4a). We then used Generalized gene-set analysis of GWAS data (as implemented in MAGMA[21]) and also identified significant enrichment of the signals confined to gene expression in cerebellar tissue (Supplementary Data 4 and Fig. 4b). The latter findings indicate that biological processes related to movement control of limbs, locomotion, adaptation of posture, and dynamic regulation of balance originated at the cerebellum could play a role shaping the complex mechanisms underlying fall risk.

**Mendelian randomization**. We then tested whether the association between seven risk factors and falls was causal by using genetic factors as instruments within a two-sample Mendelian randomization (MR) approach. Although there are many determinants and conditions that influence the risk of falling, well-powered GWASs allowing the use of adequate genetic instruments were available for alcohol consumption[22], alcohol dependence[23], BMI[24], and relative hand grip strength defined as the average of measurements of the right and left hand divided by weight[25]. In addition, we evaluated the effect of antihypertensive medication use on fall risk using genetic variants mapping to target genes for several antihypertensive drugs. For this purpose, we used genetic instruments for antihypertensive drug use that were recently created by Gill et al.[26]. These antihypertensive drug

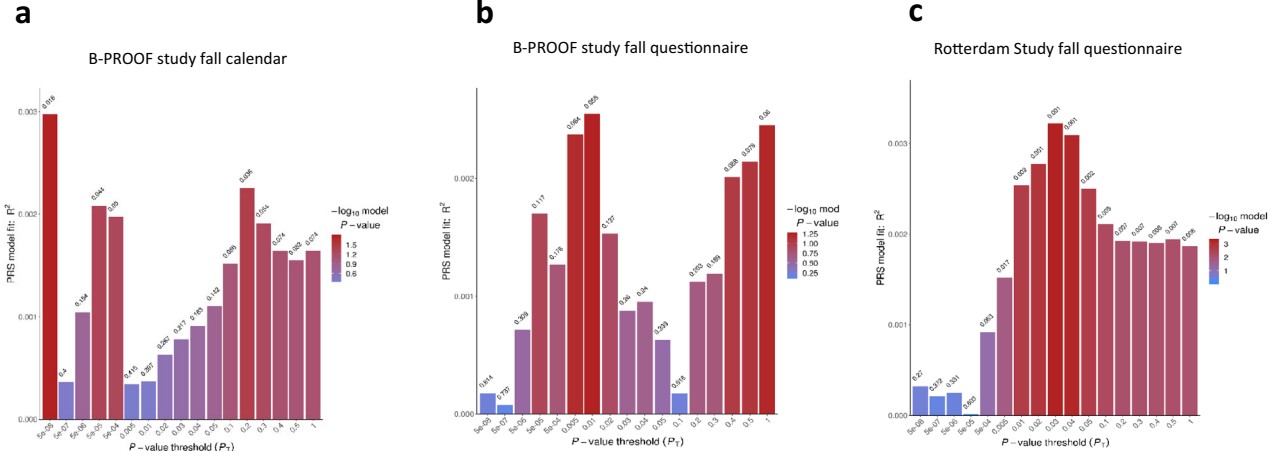

**Fig. 2 Association of falls PRS adjusted for age and sex across several different *p*-value thresholds (*x*-axis) within two different populations.**
**a** B-PROOF study falls calendar; **b** B-PROOF Study falls questionnaire; **c** Rotterdam Study falls questionnaire. The number on top of the bars represent *p*-values of the association between the score and fall risk. Color scale: −log$_{10}$ of *P*-value.

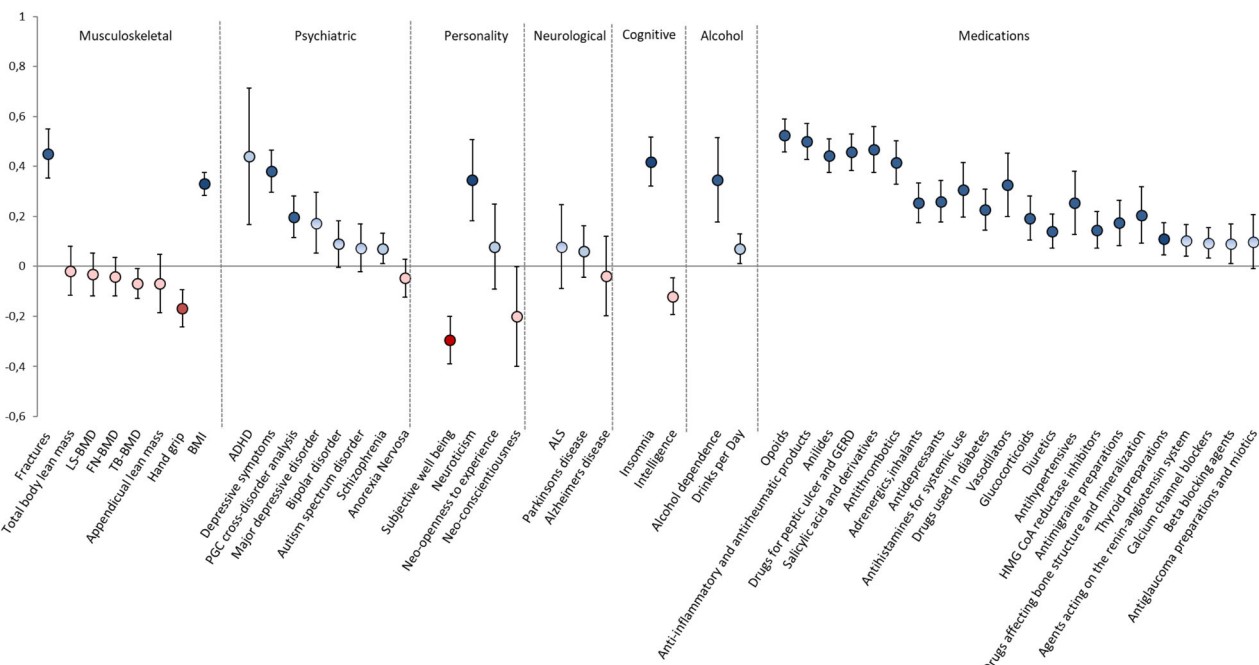

**Fig. 3 Estimates of the genetic correlation between falls and different traits and medications.** The bars around the point estimates represent confidence intervals. The blue colored point estimates indicate a positive, whereas the red colored point estimates a negative genetic correlation. The lighter shades are indication of nonsignificant correlation after correcting for multiple testing ($p = 0.05/49 = 0.001$).

medications showed no significant evidence for a causal effect. Alcohol dependence was nominally associated with increased fall risk (OR = 1.04, 95% confidence interval (95% CI) = 1.01 to 1.08, $P = 0.03$) but not after Bonferroni correction. Alcohol consumption also showed no significant evidence for a causal effect on falls. We did find evidence for a causal effect of BMI (OR = 1.13, 95% CI = 1.06 to 1.20, $P < 0.0001$) and relative hand grip strength (OR = 0.41, 95% CI = 0.23 to 0.41, $P < 0.0001$) on fall risk (Supplementary Table 2), suggesting that interventions targeted at improving muscle function and weight control may be successful at decreasing falling risk. The MR-Egger estimate for hand grip differ from the inverse-variance weighted (IVW) and WM method. Nevertheless, the $I^2$ of the model was 7.8%, indicating that the MR estimates may suffer from weak instrument bias and need to be interpreted with caution[27].

## Discussion

To our knowledge, this is the first GWAS for fall risk performed to date. Our findings indicate that fall risk is an extremely heterogeneous polygenic trait with large environmental influence. Despite such complex genetic architecture, we were able to identify variants in three loci mapping to chromosomes 5q21.2, 7p21.3, and 19q12. PRSs explained small proportion of the falls variance in two independent population-based settings. On aggregate, associated markers show significant enrichment for genes expressed in cerebellar tissue providing insight into potential mechanisms mediating fall risk. Shared genetic variation with fracture risk, muscle strength, medication use, and other risk factors suggests potential pleiotropic relationships and common biological pathways could mediate diverse aspects of falling risk.

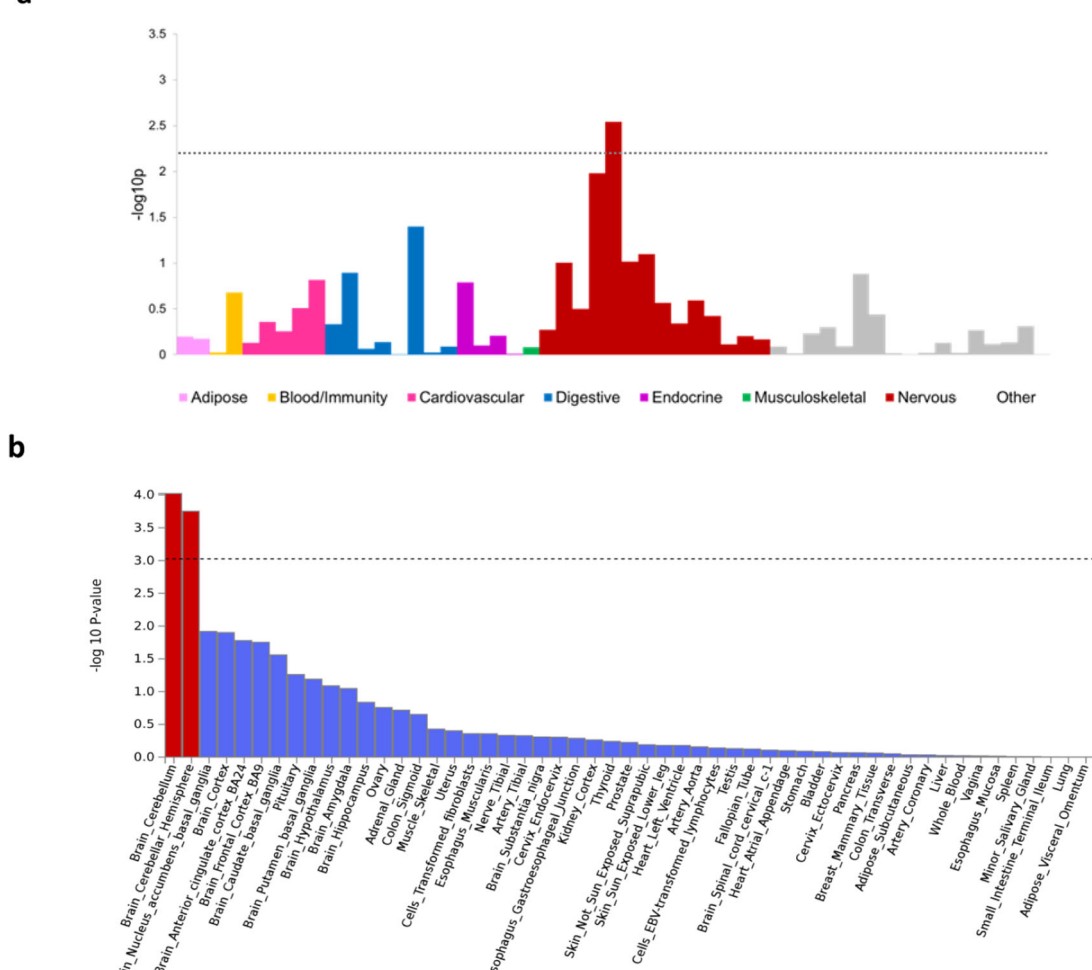

**Fig. 4 GTEx brain tissues were positively enriched for fall-associated variants. a** Heritability partitioning enrichment estimates across tissue groups using LD-SEG. **b** Tissue enrichment analysis using MAGMA. The most enriched brain tissue is the cerebellum.

Overall, falls are multifactorial in origin and many different pathways can contribute to the individual propensity to fall. Given the polygenic nature of fall risk, it is expected that a large number of genes influence the risk of falling, each with a very small contribution. Therefore, large-scale analysis such as the present UK Biobank study, are required to discover such real, but weak genetic associations. In addition, the low heritability of fall risk in our study indicates that there is a strong environmental component underlying the risk of falling. On the other hand, in twin studies, 35% of the variability in the likelihood of at least one incident fall and 45% of the variability in the risk for recurrent falls was attributed to genetic factors[13]. Notably, it has been shown that heritability may not be constant during the lifespan[28], with heritability typically decreasing with increasing age as a consequence of accumulation of environmental influences with aging[29].

We demonstrated that risk for falling has a strong positive genetic correlation with fracture and low grip strength, whereas no significant correlation was observed with BMD or lean mass. This finding implicates a mechanistic pathway influencing fracture risk that is independent of BMD. Fracture occurs when the force applied to a bone is greater than the overall bone strength. Low BMD is a key component of bone fragility, i.e., a necessary but not sufficient cause of fractures[30]. Although falls may be an independent predictor of fractures, the fall-related fracture risk dramatically increases in the presence of low BMD. However, low

BMD alone explains less than one-half of all non-vertebral fractures[31,32] and fractures occurring at higher BMD thresholds require the presence of other risk factors. Recent studies have postulated that falls and not osteoporosis per se constitute the strongest risk factor for fractures in extremely old individuals[33,34]. Therefore, falling may be a major contributing factor to the overall fracture occurrence independent of, and in addition to, age and BMD[35]. The strong genetic correlation of falls with fractures but not BMD corroborates the findings of epidemiological studies, while it also provides novel insights into the complex interplay between these traits.

The GWAS signal on chromosome 19q12 (rs2111530; MAF = 0.39) maps in the vicinity (50.6 kb) of *TSHZ3*, gene encoding a zinc-finger transcription factor involved in diverse developmental processes. Recently, a GWAS meta-analysis found a variant (rs6510186) near this gene associated with total body BMD (TB-BMD) exclusively in middle-aged adults (45–60 years old)[36]. Nevertheless, this variant is unlikely to arise from the same association signal, as it is not in LD with the top variant from our GWAS (distance 236.4 kb, $r^2 = 0.003$). There is scarce information about the function of the *TSHZ3* gene, except that it is suggested to be part of a node of 24 genes with high degree of connectivity (i.e., hub genes) with strong levels of expression in early fetal cerebral cortical development[37,38]. Reduced expression of *TSHZ3* resulting in caspase upregulation has been proposed to be involved in the pathogenesis of Alzheimer's disease[39].

Moreover, genetic linkage[40,41] and association studies[42] have identified this gene as a potential candidate for autism susceptibility disorders. Altogether, this could suggest a role of TSHZ3 in cortical development and in the pathogenesis of neurodevelopmental disorders. Pending additional evidence of its involvement in cerebellar biology, our findings suggest its plausible involvement in susceptibility to fall. Annotations relevant to TSHZ3 include both eQTLs and chromatin interactions, which could further support evidence for the gene involvement in falling susceptibility. The 7p21.3 variant (rs2709062; MAF = 0.50) is also intergenic located 127.1 kb upstream of PER4, a non-annotated pseudogene (PER3s), affiliated with a lncRNA, and of which little is known about its function. Variants in PER1, PER2, and PER3 (involved in circadian rhythm regulation) have all been identified as associated with human stature[43] and PER1 with estimated BMD[44], among other traits. Further elucidation of regulatory elements acting through PER4 or other genes will be crucial to establish its potential relation to falls. Next, the combined meta-analysis of all participating cohorts yielded another signal (rs2431108 MAF = 0.33) just surpassing the GWS threshold ($P = 4 \times 10^{-8}$). Therefore, the likelihood of a false positive cannot be excluded until additional evidence of replication becomes available, implicating robustly this locus with fall susceptibility. The SNP maps to 5q21.2 in RP11-6N13.1, a long intervening/intergenic noncoding RNA, which does not overlap protein-coding genes. This SNP have been previously reported in association with several psychiatric traits such as insomnia, anxiety, neuroticism, and depression. Overall, none of the lead SNPs show any evidence for significant eQTLs and, given the lack of information, we cannot claim if the closest genes are also the causal genes.

The PRS analyses performed in two independent prospective cohorts corroborated the polygenic architecture underlying fall risk. The joint effect of PRS could reliably determine some of the variation underlying fall risk, implicating variants associated up to a significance level of $5 \times 10^{-3}$. The scores constructed from the two most significantly associated variants were more strongly associated when using the more sensitive prospective falling assessment (fall calendars, used in the B-PROOF study), whereas employing a retrospective definition of falling (as used in the discovery and in both replication studies) resulted in the strongest PRSs arising from the inclusion of variants below the genome-wide significant level. These discrepancies can be the consequence of many different factors, including differences in assessment methodology, as retrospective falls were self-reported and may underestimate the occurrence of falls as a result of recall bias[45]. On the other hand, fall calendars are considered the best tool available for reliable fall assessment in older adults, providing more accurate information on falls (i.e., prospective assessment) as participants report fall incidents each week. Nevertheless, the PRS results should be interpreted with caution given the low SNP heritability of fall risk[46].

Medication use is recognized as an important risk factor for falling, whereas polypharmacy among the elderly has increased dramatically in the past decades[47]. Epidemiologically, several types of psychotropic, cardiac, and analgesic drugs are associated with falls, typically including sedatives and hypnotics such as benzodiazepines, antipsychotics, antidepressants, diuretics, antiepileptics, and opioids[48–50]. In line with the epidemiological relationship, we found that falls had significant genetic correlation with the use of most of these medication categories and in the expected directions. The strongest genetic correlation was observed with opioids and anti-inflammatory drug use. These findings suggest that some of the genetic predisposition for falling risk is shared with the use of medication associated with falling risk. Yet, the causal mechanistic pathways can differ across different pleiotropic (vertical vs. horizontal) relationships. Vertical pleiotropy will be relevant for medications causally related to fall risk, where the genetic correlation with a condition will also be related to the drug indication all together, pointing to a common biological pathway influencing all three components (fall risk, condition, and medication). Another form of vertical pleiotropy will be expected to arise when the medication is not necessarily causally related to fall risk (e.g., NSAIDs), but the condition driving the drug indication (e.g., arthrosis or other musculoskeletal disorder affecting mobility) will be the causal factor leading to falling. True horizontal pleiotropy is much more difficult to ascertain in a largely polygenic trait[51] such as falling risk, but we can expect it to arise with specific conditions through biological pathways influencing neurological/cerebellar function as further discussed below.

Reduced hand grip strength and increased BMI are risk factors identified by MR to be causally related to fall risk. A decrease in relative hand grip strength was observed to be causally associated with an increased risk of falling. Therefore, muscle weakness is a clinically relevant risk factor for falls that should be assessed and treated in older adults at risk for falls[52]. Our MR results also support the evidence from recent observational studies that older obese individuals have greater risk of falling[53,54]. The exact mechanisms by which BMI increases the risk of falling in older adults remain unclear and require further exploration. One possible explanation is that obesity may alter balance control, which is an important risk factor for falling[55].

Further, using grouped cell type and tissue expression analysis, the cerebellum was the most significant enriched tissue. The cerebellum plays an important role in motor control and maintaining postural balance[56]. Cerebellar disorders can lead to orthostatic hypotension, vertigo, and syncope, all important risk factors for falls[57,58]. All these factors can increase the fall risk, either individually or combined.

Some limitations of our study need to be noted. Although self-reported measures can appear robust for other phenotypes such as birthweight[59], it is possible that recall bias may have influenced the assessment of fall risk. Falling is a complex, heterogeneous trait and, in most of the cases, it may be attributed to non-genetic factors (e.g., medication use, mobility disorders, and hazardous household environments). Also, given the UKBB age range (40–69 years) and inclusion of European ancestry individuals only, we cannot assume that our results generalize to other age groups in whom falls are more common, and/or ancestral populations. There are other factors explaining the large heterogeneity underlying fall risk. Different fall patterns are observed between young and old people, whereas on the other hand, the number of co-morbid conditions and medication use associated with falls also increase with age[60]. The genetic susceptibility for falls may be of less importance in individuals with multiple fall risk factors regardless of age. Different factors can increase the risk of falling across different age decades. Next, the heritability and genetic susceptibility of fall risk might be higher in individuals with recurrent falls, which we were not able to test in the current study. Similarly, a GWAS on medication use-related falls is warranted to address the causal role of medication and understand further the biological pathways underlying fall risk. Moreover, participants from the UK Biobank were included in the GWASs of both relative hand grip strength and falls. This sample overlap might increase the probability of a Type I error resulting in false-positive findings; thus, it needs to be replicated in independent efforts. There is an ongoing collection of hospital admission data in the UK Biobank that may provide an improved and well-powered resource for future research. Lastly, in our GWAS study, we only tested SNPs for association with falling risk. We did not consider other forms of genetic variation such as structural changes, i.e., copy number variations, in/dels or

inversions, which may also contribute to the genetic landscape of falling risk. Similarly, we also did not test for potential epigenetic modifications.

In conclusion, our study demonstrated that fall risk is a heritable, heterogeneous, and polygenic trait genetically correlated with fracture risk and grip strength, among other neuropsychiatric and medication traits. The cerebellum tissue enrichment of fall-associated variants supports the mediation of postural balance in the etiology of falls. Our study provides novel biological insight that can be used for optimizing strategies directed at preventing falls and their associated deleterious consequences in aging individuals.

## Methods

**Study population**. Our analyses were performed using data from the UK Biobank study. Briefly, the UK Biobank is a large prospective cohort study of approximately a half-million adult (ages 40–69 years) participants living in the United Kingdom, recruited from 22 centers across the United Kingdom in 2006–2010[61]. We use a subsample of the total study, who were identified as or white European ancestry using a combination of genetic principal components and self-reported ethnicity. Ethical approval was granted by the Northwest Multi-centre Research Ethics Committee and written informed consent was obtained from all participants.

**Assessment of falls**. The number of falls in the UK Biobank was self-reported via a touch screen questionnaire. In total, 89,076 individuals have reported that they have had one or more falls answering the question "In the last year have you had any falls?" Individuals who selected "prefer not to answer" or "do not know" were set to missing; the rest of the population were classified as controls ($N = 362,103$).

**GWAS data and imputation**. The majority of UK Biobank participants were genotyped with the Affymetrix UK Biobank Axiom Array (Santa Clara, CA, USA), whereas 10% of participants were genotyped with the Affymetrix UK BiLEVE Axiom Array. Imputation was performed using the Haplotype Reference Consortium (HRC) panel[62] and was carried out with the IMPUTE2 software. Detailed quality control and imputation procedures are described elsewhere[63]. Only participants of white European ancestry (identified using a $k$-means clustering approach based on genetic principal components, as well as self-identification) were analyzed.

**Association analysis**. Genetic association analyses were performed using BOLT-LMM[64]. Briefly, this method uses a linear mixed model to account for relatedness and population structure using a relationship matrix. Fall risk was corrected for age and sex in logistic regression models. SNP association was tested for all autosomal variants. Individuals were excluded based on unusually high heterozygosity or >5% missing genotype rate, a mismatch between self-reported and genetically inferred sex. SNP exclusions were made based on low MAF (<1%) and low imputation quality (info < 0.3). SNPs with $P \leq 5 \times 10^{-7}$ were considered suggestive, whereas SNPs with $P \leq 5 \times 10^{-8}$ were considered GWS.

**Replication and meta-analysis**. Suggestive SNPs were selected based on an arbitrary $P$-value threshold of $\leq 5 \times 10^{-7}$ and were later followed for replication in smaller and older prospective population-based studies, i.e., the Rotterdam Study (1009 cases and 4925 controls) and B-PROOF (1206 cases and 1364 controls) cohorts. The Rotterdam Study is an ongoing population-based cohort within a suburb in Rotterdam. Its design, objective, and methods have been described in detail[16]. Briefly, the study was initiated in 1989 and 7983 participants aged 55 years and above were included. Participants were interviewed and underwent an extensive set of examination that were repeated every 4–5 years. B-PROOF has been also described in detail elsewhere[15]. In short, it is a multi-center, randomized, placebo-controlled, double-blinded trial investigating the efficacy of vitamin B and folic acid supplementation on the prevention of fractures in people aged 65 years and older. In total, 2919 participants were included and followed for 2–3 years. Both studies have been approved by the medical ethics committee. In the Rotterdam Study, fall history was assessed from baseline questionnaire. In B-PROOF, retrospective and prospective falls were reported. Prevalent falls were assessed using a baseline questionnaire, whereas fall incidences during follow-up were reported prospectively using a falls' calendar in a period of 2–3 years. The baseline questionnaire in both cohorts consisted of a single question: "Have you fallen in the past 12 months?" In B-PROOF, we used fall incidence for the GWAS analysis, as it provided larger sample size (500 participants had missing information on prevalent falls). Both studies used commercially available genome-wide arrays to genotype their participants. SNPs were imputed to the HRC reference panel[62] (build 37) using the Michigan Imputation Server.

**Gene-based testing and functional mapping**. Gene-based GWAS analysis was carried out with MAGMA 1.6[21] using the default settings implemented in FUMA[65]. The analysis was performed using the falls GWAS summary statistics as input file where each SNPs were assigned to a gene using the NCBI 37.3 gene definition. First, each individual SNP in a gene was tested separately and SNP $p$-values within a gene were aggregated to derive a gene test statistic[21]. According to the number of tested genes, the level of gene-wide significance was set at 0.05/ $18,615 = 2.7 \times 10^{-6}$. The gene-base $p$-values were then utilized to perform gene-set enrichment analyses using MAGMA (implemented in FUMA); where gene expression values for 53 specific tissue types were obtained from GTEx[66]. The gene-set enrichment analysis estimated the association between highly expressed genes in a specific tissue and genetic associations from the gene-based analysis. Functional annotation (i.e., prioritization, annotation, and interpretation) of GWAS results was also performed using FUMA[65].

**Polygenic risk scores**. We used imputed genotype data from the B-PROOF and the Rotterdam Study cohorts to calculate PRSs for each participant using the PRsice software[18]. For the construction of the PRS, we first selected SNPs with $P <$ 0.05, MAF > 0.05, and imputation quality > 0.3 from the UK Biobank GWAS. Next, we excluded SNPs with MAF < 0.05 and imputation quality < 0.3 in the Rotterdam study and B-PROOF. In total, 436,130 SNPs passed these thresholds and were followed in the PRS analysis. Seventeen PRS sets were created for a series of $P$-value thresholds ranging from $5 \times 10^{-8}$ to 1. SNPs were pruned with PLINK (version 1.9) using stringent clumping thresholds based on both linkage disequilibrium and distance using an $r^2$ of 0.05 and a distance of 300 kb. We tested each of the 17 PRSs in relation to fall risk adjusted for age and sex stratified by cohort using logistic regression models. In the B-PROOF study, we tested the scores with both prevalent and incident falls. We reported the proportion of variance explained (based on $R^2$) by each fall PRS.

**Linkage disequilibrium score regression**. *SNP-based heritability and genomic inflation*. To estimate the genomic inflation in the data and the SNP-based heritability of falls, we used LDSR[67]. The LDSR intercept provides estimate of inflation due to population stratification or model misspecification; importantly, it is not unduly affected by polygenicity[68]. On the other hand, the LDSR slope provides estimate of the heritability explained by all SNPs[67]. The analyses were restricted to HapMap3 SNPs with MAF > 5% in the 1000 Genomes European reference population. Finally, we used pre-calculated LD scores from the same reference population. (https://data.broadinstitute.org/alkesgroup/ LDSCORE/).

*Shared genetic architecture of falls and other traits*. To estimate the genetic correlation between falls and other complex traits and diseases, we used (cross-trait) LDSR[19] as implemented in the online web utility LDHub[69]. This method uses the cross-products of summary test statistics from two GWASs and regresses them against a measure of how much variation each SNP tags (its LD score)[70]. The data base of the LDHub web utility contains a range of data relating to genetic effects on common diseases and phenotypes. From the variety of traits available on LD-hub, we selected 17 cognitive, personality, psychiatric, and neurological traits/disorders a priori that can affect the function of the motor and sensory systems. These traits are essential for the planning and execution of the everyday movements and for the control of posture and balance. Further, we selected seven additional musculoskeletal traits (including fractures; site-specific BMD of the femoral neck and lumbar spine and TB; appendicular and total lean mass; and hand grip muscle strength to better understand the relationship between falling risk and the risk of fracture). In addition, locally, we estimated the genetic correlation between fall risk and 22 medication classes from the latest GWAS on medication use[20], where medications were self-reported and classified using the Anatomical Therapeutic Chemical Classification System into 1752 categories (with minimum 10 users) after careful evaluation of the medication record data. We accounted for multiple testing by using a conservative Bonferroni correction for 46 tests (0.05/46 = 0.001). Finally, we also included in the LDSR analyses BMI, alcohol consumption, and alcohol dependence tested for causal effects on falling risk using the MR approach.

*Stratified heritability and functional enrichment and tissue specificity analyses of fall-associated variants*. To identify tissues and cell types that are likely to be involved in falling susceptibility, we applied LDSC-SEG[71]. Annotation data were obtained from the LD score website (https://github.com/bulik/ldsc). For each tissue, we ranked genes by a $t$-statistic for differential expression, using sex and age as covariates, and excluding all samples in related tissues[71]. For example, we compared expression in cerebellum samples to expression in all non-brain samples. We used the top 10% of genes by this ranking and used stratified LDSR to estimate the contribution of genomic annotations to per-SNP falls heritability, adjusting for 24 main annotations categories (e.g., coding, untranslated region (UTR), promotor, and intron) in the baseline model[72]. In addition, we performed MAGMA Tissue Expression analysis (using FUMA) to test relationships between tissue-specific gene expression and disease–gene associations. MAGMA was performed using the result of gene-set analysis (gene-based $P$-value) and tested for one side (greater) with conditioning on average expression across all tissue types.

**Mendelian randomization**. We used the largest previously published GWAS meta-analyses of the traits included in the MR analyses. The construction of genetic instruments for antihypertensive drug targets are explained in details by Gill et al.[26]. To reduce potential bias due to population stratification, we restricted the analyses to studies with participants of European descent. In addition, instrumental variables which were nominally ($p < 0.05$) associated with fall risk were excluded from the analysis. The resulting individual SNP effect estimates were pooled using IVW meta-analysis. We applied a conservative Bonferroni corrected threshold to account for the multiple testing (i.e., $\alpha = 0.007$, because seven exposures were assessed). To test the third assumption (a lack of pleiotropic effects of the SNPs on the outcome, independent of the exposure), we used MR-Egger regression. Moreover, as sensitivity analyses for robust causal inference, we additionally performed MR analyses using a weighted median estimator. The analyses were conducted with the R-package MendelianRandomization[73].

**Reporting summary**. Further information on research design is available in the Nature Research Reporting Summary linked to this article.

## Data availability

The data analyzed in the current study were provided by the UK Biobank Study (www.ukbiobank.ac.uk) under UK Biobank application number 24268. The datasets generated during the current study are available at the GEFOS website http://www.gefos.org/. The data source for Fig. 1 was the falls GWAS summary statistics, which are available at the GEFOS website http://www.gefos.org/. Additional datasets that have been used for analysis can be found on the links to their corresponding reference papers and URLs (see below).

## Code availability

All custom codes used in this study are freely available online in the referenced articles and the following URLs:

FUMA GWAS http://fuma.ctglab.nl
LDSC (LD SCore) https://github.com/bulik/ldsc
LD Hub http://ldsc.broadinstitute.org/ldhub
GTEx Portal https://www.gtexportal.org/home/
PRSice-2: Polygenic Risk Score software https://www.prsice.info/

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

## Acknowledgements

This research has been conducted using the UK Biobank Resource (Application number 24268). The Rotterdam Study was supported by The Netherlands Organization for Health Research and Development (ZonMw), The Netherlands Organisation for Scientific Research (NWO), the Ministry of Health, Welfare, and Sport. In addition, the Netherlands Organization for Health Research and Development supported authors of this manuscript (ZonMw VIDI 016.136.367 to K.T., F.R., and C.M.G.).

## Author contributions

Design the study: K.T., F.R.D., J.R.B.P., and F.R. Analyzed the data: K.T., L.J.S., F.R.D., J.R.B.P., and F.R. Wrote first draft of the paper: K.T., F.R.D., C.M.G., J.R.B.P., and F.R. Contributed and approved the final version of the paper: K.T., L.J.S., F.R.D., J.R.B.P., F.R., C.M.G., Y.-H.H., S.Z., N.M.v.S., L.C.P.G.M.d.G., N.v.d.V., J.B.R., D.K., D.P.K., and A.G.U.

## Competing interests

The authors declare no competing interests.
