## [Peer Review File · Communications Biology]

Reviewers' comments:

Reviewer #1 (Remarks to the Author):

This is a very well conducted GWAS analysis that focuses on falls data in the UK Biobank cohort. The genetic correlations between falls and fracture risk but not bone mineral density are particularly interesting, as is the enrichment for genes expressed in the cerebellum. I have a few fairly minor comments and suggestions:

1. Is it fair to call falls a polygenic trait when the PRS with the "more sensitive prospective" assessment performs best with just genome-wide SNPs? Also, for the PRS section, is there any reason why you didn't consider a score that used all SNPs?
2. Was a multiple comparison test performed for the genetic correlation analysis? Correlations around 0.4 are not (in my opinion) "strongly positive" nor are correlations of -0.12 moderate. They are moderate and small, respectively. (Top of page 5)
3. Why were $5e-6$ and $5e-7$ selected as suggestive thresholds? Are these arbitrary?
4. I think it might be nice to look at the phenotypic associations between medication and falls. I think this should be possible as all data stem from the UK Biobank study. Given that we don't have time of event for falls, I wonder if strong conclusions can be made in terms of the temporal associations. There was no evidence from the MR to suggest a causal effect of medication of falls.
5. Please could you clarify that the MR estimates are per SD - this is mentioned in Table S1, though there is a typo in the footnote pre -> per. Also, I am not a MR expert but the Egger estimates seem to suggest that there is no evidence for a causal association between grip strength and falls.
6. Discussion, "providing insight into mechanisms mediating fall risk". Maybe temper this to "insight into potential mechanisms"
7. Did you control for assessment centre and genotyping batch in the GWAS?

Typos:

Page 1: "Genetic varia[n]tion". Do the authors think it is likely that fall risk stratification can or will be implemented?

Page 1: "female[s] carriers of the/an ACTN3...."

Page 5: "an[d] also identified significant enrichment of the signals..."

Page 12: "can appear[ed] robust"

Page 13: "false-positive flinging" I assume this is a typo? It's not an expression I've come across before!

Page 13: "Princip[al] components analysis"

Reviewer #2 (Remarks to the Author):

Thank you for the opportunity to review the manuscript "Genetic basis of falling risk susceptibility"

by Trajanoska et al.

Overall, I find the work comprehensive and commendable. The analysis is neat and well-presented. I recommend this manuscript for publication in the journal. However, I request the authors to consider the following points in the final revision -

1. For Table 1 footnote, include column descriptions e.g. EA effect allele, NEA non-effect allele, EAF effect allele frequency.

2. The SNP datasets used were generated by Affymetrix array that queries only about ~800,000 positions. While it can obviously be useful for such SNP based analysis, it can not explain the entire "genetics" on its own. Variants at other positions (WGS), structural variants, CNVs, epigenetics could also potentially explain the variability. I'd suggest authors make a comment about it in the Discussion section, under limitations.

3. There is no mention of Supplementary figure 3b in text.

4. Abstract, line 43: "The analysis revealed a small, but significant SNP-based heritability (2.7%)". It is unclear from the analysis where is "2.7%" derived from?

5. Typos :

Line 167: "We then used Generalized gene-set analysis of GWAS data (as implemented in MAGMA18) an also identified ...".

Line 465: "The resulting individual SNP effect estimates were using pooled ..."

Reviewer #3 (Remarks to the Author):

The paper by Trajanoska et al. reports the results from a GWAS on fall susceptibility, which consists of a discovery (UK Biobank) and replication phase (mostly based on PRS in two independent cohorts). The authors identified 3 novel loci for fall susceptibility that were genome-wide significant after combining the discovery and replication phase results. In addition, they showed that the trait is genetically correlated with several traits, including fracture risk and muscle strength. They also perform some follow-up analyses (including enrichment analysis and Mendelian Randomization) to look for underlying processes and to study the causal relationships with (some of) the genetically correlated phenotypes.

I think the manuscript is well written (with some small spelling errors), the statistical analyses are solid and the results are interesting for the field. Hence, I only have some relatively minor comments.

- In the Abstract the authors mention that the SNP-based heritability for fall susceptibility is 2.7%, but I could not find these results anywhere in the manuscript. Hence, the heritability estimate should thus be added to the Results section of the manuscript (including a description of the calculation in the Methods section).

- In the Methods section the authors mention that they removed SNPs with a low imputation quality ($info < 0.4$) in the UK Biobank. However, in the results section they mentioned that they used an imputation quality > 0.3 . Hence, this does not match and should thus be corrected.

- I could not find a QQ-plot of the discovery GWAS in UK BioBank in the manuscript. This should

be added as a Supplementary Figure.

- It would be nice if the authors could also make a Circos plot for the locus on 5q21.3, given that this locus also reaches genome-wide significance after the replication phase.

- Although I have limited knowledge about polygenic risk score (PRS) analyses, it does not seem right to use PRS based on SNPs with P-values $>5 \times 10^{-8}$ or maximally $>5 \times 10^{-7}$, given that this would result in inclusion of many SNPs with potential false positive associations. Hence, the authors should clarify why they used PRS based on lower thresholds.

- The estimates for the genetic correlation of fall susceptibility with fracture and muscle strength mentioned in the text ($r_g = 0.35$ and $r_g = -0.24$, respectively) do not seem to match with what is provided in Figure 3 ($r_g \sim 0.45$ and $r_g \sim 0.15$, respectively). Hence, this should be corrected. Moreover, it would be interesting to know the genetic correlations with alcohol consumption, alcohol dependence and BMI (i.e. these could be included in this Figure), given that the authors used these phenotypes for their Mendelian Randomization analyses.

- In the Discussion section the authors should mention the lack of evidence for eQTL effects for SNPs in the loci on 7p21.3 and 5q21.3, to indicate that it is not yet clear which genes at these loci are implicated in the phenotype (there is currently no mention of this). Moreover, it would be interesting to see the results of a look-up of the lead SNPs at the identified loci in Phenoscanner (<http://www.phenoscanner.medschl.cam.ac.uk/>).

Joris Deelen

Reviewer #4 (Remarks to the Author):

It is rather disappointing that you find only two-three loci affecting risk of falling after analyzing more than 90 thousand cases and hundreds of thousands of controls.

I found the manuscript somewhat confusing with regard to the questions of discovery and replication.

Please decide if you want to claim two or three loci "genome-wide significant", and if you have replication set or not. According to the text lines 94-96 and Table 2 (see the discovery part, UK Biobank), there should be two loci. If you want to claim significance as significance after meta-analysis (then you will have at least three loci, as indicated in the abstract, line 197, and Table 1 combined data), I would advise you to do a genome-wide meta-analysis (but then do not use the RS and B-PROOF data set for "replication").

It is also not quite clear what you have replicated, and what is significance and interpretation of your replication. Quite clearly, you do not see replication of the two loci that were genome-wide significant in discovery. Looking into results of replication of polygenic score (Figure 2) at $5e-7$, I also do not see anything significant. In abstract, you say that "Polygenic risk scores were replicated"; do you refer to the fact that in RS, for PRS derived at p-thresholds 0.01, 0.02 and 0.03 the p-value for association between PRS and the trait is $33 \times 0.001 = 0.033 < 0.05$ (11 thresholds, three tests)? This is not quite clear from the text.

What is interpretation of this finding? That some of the (tens of thousand?) of SNPs that are identified by your GWAS at nominal $p < 0.03$ are indeed related to the fall risk, because a score made of these is associated with $p = 0.001$ in one of the two other studies?

It may be interesting to follow a bit more on significant loci you have identified, eg look for pleiotropic effects across other complex trait and "omics" GWAS studies.

Minor comments:

Line 73, please fix "variantion"

In discussion you say (lines 198-199) that "Polygenic risk scores were associated with falling risk in two independent population-based settings"; looking into Figure 2, I see that only results in RS, but not in B-PROOF, pass the multiple testing correction. So, the two "independent population-based settings" mean the discovery cohort of UKB and one of the replication cohorts, the RS? I find this somewhat confusing.

Reviewers' comments:

Reviewer #1 (Remarks to the Author):

This is a very well conducted GWAS analysis that focuses on falls data in the UK Biobank cohort. The genetic correlations between falls and fracture risk but not bone mineral density are particularly interesting, as is the enrichment for genes expressed in the cerebellum. I have a few fairly minor comments and suggestions:

Dear Reviewer #1, We thank you for the constructive comments which have helped us improve our manuscript. We address below each comment point-by-point to the best of our possibilities.

1. Is it fair to call falls a polygenic trait when the PRS with the "more sensitive prospective" assessment performs best with just genome-wide SNPs? Also, for the PRS section, is there any reason why you didn't consider a score that used all SNPs?

Reply: We acknowledge that the different behaviour between using only the genome-wide significant (GWS) SNPs or SNPs associated at lower significance (higher p-value) thresholds in the score demands further explanation when applied to different phenotype definitions. Indeed, the score composed of GWS SNPs performed the best when falling risk was more robustly assessed (fall-calendar in B-Proof); while the score using SNPs was seen associated at higher significance thresholds, performed better when falls was defined based on questionnaire data (Rotterdam Study). Therefore, the type of fall ascertainment influenced the associations but is unlikely to shape genetic architecture. We stand by our statement that fall risk (independent of how it is ascertained) is a polygenic trait; in this context, shaped by multiple (hundreds) genetic variants and under the influence of environmental factors, both contributing to the underlying trait heterogeneity.

For the second point, we tested a score using all SNPs in the Rotterdam Study where we observed that the variance explained by SNPs plateaued with P-values > 0.05 (**Figure A**). Therefore, we restricted our analysis to 0.05 in BPROOF as well. Now, we re-ran the PRS analysis using additional P-value thresholds (p_i) ranging from 5×10^{-8} to 1 in the BPROOF study as well. From **Figure B** and **C**, we can conclude that SNPs with P-value < 0.05 explain the highest proportion of falls variance. For falls calendar $p_i < 5 \times 10^{-8}$ was the best fit, whereas for fall questionnaire $p_i < 0.01$ was the best fit. In B-proof, the variance explained by both scores dropped to then increase after the p-value threshold of 0.05, again supporting the polygenic nature of falls, since increased variance is still observed with markers genome-wide despite the low signal to noise ratio. This drop was not observed in the Rotterdam study as it had more power at higher significance thresholds, where less SNPs are contributing to variance explained. We have included the new figures within the main manuscript and updated the methods information.

A. Rotterdam study fall questionnaire

B. B-PROOF study fall calendar

C. B-PROOF study fall questionnaire

2. Was a multiple comparison test performed for the genetic correlation analysis? Correlations around 0.4 are not (in my opinion) "strongly positive" nor are correlations of -0.12 moderate. They are moderate and small, respectively. (Top of page 5)

Reply: We agree with the reviewer that multiple testing correction should be in place and also that the classification of the correlations should be used according to convention. Initially, the genetic correlations were not corrected for multiple testing. As shown below, after correcting the genetic correlation analysis for multiple testing the majority of the traits in **Figure 1** passed the Bonferroni adjusted p-value threshold ($p=0.05/46=0.001$). We have added the following sentence in the method section (**Page 19**):

"We accounted for multiple testing by using a conservative Bonferroni correction for 46 tests ($0.05/49=0.001$)."

In addition, we expanded the descriptive caption of Figure 3 to read as:

"The lighter shades are indication of non-significant correlation after correcting for multiple testing ($p=0.05/49=0.001$)."

For the second point we agree with the reviewer that 0.4 can be not labelled "strong" nor -0.12 as "moderate" correlation. The statement was inappropriately used referring to the magnitude of the p-values. Following the reviewer's suggestion, we have now corrected this in the text to read large, moderate and small where appropriate.

3. Why were $5e-6$ and $5e-7$ selected as suggestive thresholds? Are these arbitrary?

Reply: Yes, in general these thresholds are arbitrary but are commonly used across GWAS. We have added a sentence in the methods to clarify this on page 16:

"Suggestive SNPs were selected based on an arbitrary P value threshold of $\leq 5 \times 10^{-7}$..."

4. I think it might be nice to look at the phenotypic associations between medication and falls. I think this should be possible as all data stem from the UK Biobank study. Given that we don't have time of event for falls, I wonder if strong conclusions can be made in terms of the temporal associations. There was no evidence from the MR to suggest a causal effect of medication on falls.

Reply: We agree with the review that it would be of interest to investigate the association between medication and falls. However, we feel that a comprehensive assessment of the phenotypic associations is beyond the scope of this study, i.e., currently focused on the genetic basis of falling. Nevertheless, we provide in the table below an overview of associations between common medication use and falls, corrected for age and sex. It is obvious that there is widespread association, but it is also clear that the magnitude of many are likely to be the result of confounding or reverse causality. As the reviewer pointed out, there is also a temporal barrier to pursue these associations further, since medication use in the UK Biobank was assessed from self-reports during the time of assessment, while falls were inquired with respect to events occurring during the past year. Altogether, we have abstained from performing further analysis and from including this in the paper, which is already comprehensive.

Medication	Odds Ratio	SE	z	LCI 95%	UCI 95%
aspirin	1.268	0.013	22.750	1.242	1.294
ibuprofen	1.307	0.014	25.600	1.280	1.334
simvastatin	1.174	0.013	14.320	1.149	1.200
omeprazole	1.622	0.022	35.490	1.580	1.666
glucosamine	0.998	0.014	-0.130	0.970	1.027
bendroflumethiazide	1.134	0.017	8.390	1.101	1.167
cod_liver	0.928	0.015	-4.750	0.900	0.957
ramipril	1.209	0.020	11.490	1.171	1.249
amlodipine	1.179	0.020	9.590	1.140	1.220
levothyroxine_s	1.211	0.020	11.390	1.172	1.251
atorvastatin	1.347	0.027	15.140	1.296	1.400
lansoprazole	1.595	0.028	26.860	1.542	1.650
atenolol	1.158	0.021	8.130	1.117	1.199
fish_oil	0.918	0.018	-4.450	0.884	0.953
multivitamins	1.016	0.021	0.770	0.976	1.058
metformin	1.585	0.031	23.250	1.525	1.648
lisinopril	1.126	0.024	5.570	1.080	1.174
ventolin	1.273	0.026	11.870	1.223	1.324
cocodamol	2.241	0.045	40.480	2.155	2.330
diclofenac	1.680	0.037	23.440	1.608	1.754
amitriptyline	2.243	0.050	36.420	2.148	2.343

5. Please could you clarify that the MR estimates are per SD - this is mentioned in Table S1, though there is a typo in the footnote pre -> per. Also, I am not a MR expert but the Egger estimates seem to suggest that there is no evidence for a causal association between grip strength and falls.

Reply: We have now clarified that the effect estimates in **Supplementary table 1** are not per SD but per unit change in the exposure, i.e., hand grip= m^2 , BMI= kg/m^2 and alcohol consumption=log transformed.

Regarding the second point, indeed, the MR Egger estimates are not significant. However, the confidence intervals (CIs) of the effect estimates are quite large and contain the CIs of the effect estimates of the IVW and weighted median MR method; indicating that there is no difference in the effect estimates between the models. In general, the standard error of MR Egger effect estimate is always larger than the one from the IVW method, thus, provide less precise estimates ¹. The importance of the Egger models is in the situation when there is evidence for a significant excess of signal in one particular direction – evidenced by the p-value for the intercept in the Eggers model. In addition, the MR-Egger effect estimate is also dependent on the strength of the SNP-exposure association. If some of the instruments have stronger association with the exposure compared to the others this will impact the slope of the MR-regression. These types of influential points will not be of impact in the IVW method.

Therefore, discrepancies may arise between the MR-Egger and IVW method. In addition, Bowden et al.² suggested to use the I^2 statistic as a measure of instrument strength for the MR-Egger method where values less than 90% indicate that the MR-Egger estimate may suffer from 'weak instrument bias'. In our case the I^2 was 7.9% which could have led to dilution of the MR effect estimate. Therefore, the findings from the MR-Egger need to be interpreted with caution. We have added the following sentences to the manuscript (**Page 8**):

"The MR-Egger estimate for hand grip differs from the IVW and WM method. Nevertheless, the I^2 of the model was 7.8% indicating that the MR estimates may suffer from weak instrument bias and need to be interpreted with caution."²⁷

6. Discussion, "providing insight into mechanisms mediating fall risk". Maybe temper this to "insight into potential mechanisms"

Reply: We thank the reviewer for the comment and have now changed this sentence (**Page 8**) to read as ***"On aggregate, associated markers show significant enrichment for genes expressed in cerebellar tissue providing insight into potential mechanisms mediating fall risk"***.

7. Did you control for assessment centre and genotyping batch in the GWAS?

Reply: The GWAS analysis were not adjusted for assessment centre and genotyping batch. During the marker-based quality control markers that were not reliable across all batches was excluded from the final dataset. In addition, it has been shown genotyping batch effect to be more relevant for rare SNPs with $MAF < 0.01$ ³ which were excluded from our meta-analysis.

Typo: Page 1: "Genetic varia[n]tion". Do the authors think it is likely that fall risk stratification can or will be implemented?

Reply: Risk stratification may arise from understanding the underlying biological mechanisms and our findings do provide some insight in that direction. We are currently setting up a study which will allow us to explore whether genetic factors associated with falls have additive value to the clinical risk prediction and stratification models.

Page 1: "female[s] carriers of the/an ACTN3..." **Page 5:** "an[d] also identified significant enrichment of the signals..." **Page 12:** "can appear[ed] robust"; **Page 13:** "false-positive flinging" I assume this is a typo? It's not an expression I've come across before. **Page 13:** "Princip[al] components analysis"

Reply: We have corrected now the typos and they are highlighted in the manuscript.

Reviewer #2 (Remarks to the Author):

Thank you for the opportunity to review the manuscript "Genetic basis of falling risk susceptibility" by Trajanoska et al. Overall, I find the work comprehensive and commendable. The analysis is neat and well-presented. I recommend this manuscript for publication in the journal. However, I request the authors to consider the following points in the final revision:

Dear Reviewer #2, We thank you for the kind assessment and constructive comments which have helped us to improve our manuscript.

1. For Table 1 footnote, include column descriptions e.g. EA effect allele, NEA non-effect allele, EAF effect allele frequency.

Reply: We have now included all abbreviations from **Table 1** in the table footnote.

2. The SNP datasets used were generated by Affymetrix array that queries only about ~800,000 positions. While it can obviously be useful for such SNP based analysis, it cannot explain the entire "genetics" on its own. Variants at other positions (WGS), structural variants, CNVs, epigenetics could also potentially explain the variability. I'd suggest authors make a comment about it in the Discussion section, under limitations.

Reply: Thank you for the suggestion. We have added the following sentence in the discussion under the limitations paragraph (**page 14**): *"Lastly, in our GWAS study we only tested SNPs for association with falling risk. We did not consider other forms of genetic variation such as structural changes i.e., copy number variations (CNVs), in/dels or inversions; which may also contribute to the genetic landscape of falling risk. Similarly, we also did not test for potential epigenetic modifications"*.

3. There is no mention of Supplementary figure 3b in text.

Reply: Thank you for pointing this out. Supplementary figure 3 is now Supplementary figure 4. In order to improve the readability of the section we changed the presentation order of Supplementary Figure 4B and 4C and added the following sentence to the Results section regarding Figure 4B (now 4C) on **page 4**: *"However, only 3.3% possessed strong (regulomeDB score \leq 2) regulatory potential (Supplementary Figure 4C)."*

4. Abstract, line 43: "The analysis revealed a small, but significant SNP-based heritability (2.7%)". It is unclear from the analysis where is "2.7%" derived from?

Reply: We have now clarified the heritability estimate in the results section (**Page 6**):

"We then used LD-Score Regression (LDSR) to estimate the heritability (individual and shared) between falls and different diseases and traits¹⁹; restricting the analysis to common variants present in HapMap3 with MAF>5%." ... "As expected, the SNP-based heritability of falls was low ($h^2=0.027$, SE=0.002)."

In addition, in the methods section (**page 18**) we added the following:

"To estimate the genomic inflation in the data and the SNP-based heritability of falls, we used LD-score regression (LDSR)⁶⁵. The LD-score regression intercept provides estimate of inflation due to population stratification or model misspecification; importantly, it is not unduly affected by polygenicity⁶³. On the other hand, the LD-score regression slope provides estimate of the heritability explained by all SNPs⁶⁵. The analyses were restricted to HapMap3 SNPs with MAF>5% in the 1000 Genomes European reference population. Finally, we used pre-calculated LD-scores from the same reference population. "

5. Typos:

Line 167: "We then used Generalized gene-set analysis of GWAS data (as implemented in MAGMA18) and also identified ...". Line 465: "The resulting individual SNP effect estimates were using pooled ..."

Reply: Thank you. We have now corrected the typos in the manuscript (highlighted yellow).

Reviewer #3 (Remarks to the Author):

The paper by Trajanoska et al. reports the results from a GWAS on fall susceptibility, which consists of a discovery (UK Biobank) and replication phase (mostly based on PRS in two independent cohorts). The authors identified 3 novel loci for fall susceptibility that were genome-wide significant after combining the discovery and replication phase results. In addition, they showed that the trait is genetically correlated with several traits, including fracture risk and muscle strength. They also perform some follow-up analyses (including enrichment analysis and Mendelian Randomization) to look for underlying processes and to study the causal relationships with (some of) the genetically correlated phenotypes.

I think the manuscript is well written (with some small spelling errors), the statistical analyses are solid and the results are interesting for the field. Hence, I only have some relatively minor comments.

Reply: Dear Reviewer #3, We thank you for the constructive comments which have helped us improve our manuscript. We have meticulously reread the manuscript and corrected several typos throughout the manuscript (highlighted yellow) to the best of our capacity.

1. In the Abstract the authors mention that the SNP-based heritability for fall susceptibility is 2.7%, but I could not find these results anywhere in the manuscript. Hence, the heritability estimate should thus be added to the Results section of the manuscript (including a description of the calculation in the Methods section).

Reply: We thank the reviewer for pointing this out and apologize for omitting the heritability estimate from the results section. As addressed to Reviewer 2 in comment 4, we have corrected this to read:

We have now clarified the heritability estimate in the results section (**page 6**):

"We then used LD-Score Regression (LDSR) to estimate the heritability (individual and shared) between falls and different diseases and traits¹⁹; restricting the analysis to common variants present in HapMap3 with MAF>5%." ... "As expected, the SNP-based heritability of falls was low ($h^2=0.027$, $SE=0.002$)."

In addition, in the methods section (**page 18**) we stated:

"To estimate the genomic inflation in the data and the SNP-based heritability of falls, we used LD-score regression (LDSR).⁶⁵ The LD-score regression intercept provides estimate of inflation due to population stratification or model misspecification; importantly, it is not unduly affected by polygenicity. ⁶⁶ On the other hand, the LD-score regression slope provides estimate of the heritability explained by all SNPs. ⁶⁵ The analyses were restricted to HapMap3 SNPs with MAF>5% in the 1000 Genomes European reference population. Finally, we used pre-calculated LD-scores from the same reference population. "

2. In the Methods section the authors mention that they removed SNPs with a low imputation quality ($\text{info} < 0.4$) in the UK Biobank. However, in the results section they mentioned that they used an imputation quality > 0.3 . Hence, this does not match and should thus be corrected.

Reply: We used INFO score threshold of > 0.3 as filter and have now corrected this in the methods section (page 16).

3. I could not find a QQ-plot of the discovery GWAS in UK BioBank in the manuscript. This should be added as a Supplementary Figure.

Reply: Thank you for the suggestion. We have now added the QQ plot as **Supplementary Figure 2** and referred to it in the manuscript on page 3.

4. It would be nice if the authors could also make a Circos plot for the locus on 5q21.3, given that this locus also reaches genome-wide significance after the replication phase.

Reply: We have now added the Circos plot for the 5q21.3 locus as **Supplementary Figure 5c** and referred to it in the text on page 5:

“The new locus did not harbour any genes with relevant eQTL and/or chromatin interactions (Supplementary Figure 5c).”

Although I have limited knowledge about polygenic risk score (PRS) analyses, it does not seem right to use PRS based on SNPs with P-values $> 5 \times 10^{-8}$ or maximally $> 5 \times 10^{-7}$, given that this would result in inclusion of many SNPs with potential false positive associations. Hence, the authors should clarify why they used PRS based on lower thresholds.

Reply: The traditional PRS analysis (or so-called genetic risk score [GRS]) have been mainly focused on SNPs with P-values $< 5 \times 10^{-8}$. However, these SNPs explain small proportion of the trait variance. During the past decade growing evidence has emerged on the polygenic nature of many complex traits^{4,5,6,7}; implying they are affected by numerous genes with small individual effects⁸. In order for these SNPs to pass the stringent GWAS P-value threshold we need very large sample sizes. The stringent P-value threshold of $< 5 \times 10^{-8}$ is quite relevant when defining lead SNPs as all our post GWAS analyses utilize this set of SNPs. However, in the PRS analyses we are more interested in the genome-wide genetic landscape of the trait. Purcell et al.⁹ have shown that using more liberal P-value thresholds can actually improve the utility of PRS for highly polygenic traits. Indeed, as the P-value threshold increases, the number of SNPs included in the PRS increases and hence the ratio of false : true positives increases; but GRS can tolerate the inclusion of false-positive findings¹⁰. The PRS set with the best variance explained (R^2) from the regression model between the PRS and the trait is referred to as the most optimal PRS. Above the optimal PRS P-value threshold, the proportion of false-positives is expected to be larger. In general, we can use all SNPs across the genome into one score¹¹ as long as proper quality steps are applied. In order to clarify this for the readers we added the following in the results section (Page 5):

“Next, we evaluated the ability of polygenic risk scores (PRSs) constructed from the UK Biobank GWAS results to discriminate between fallers and non-fallers in two independent prospective cohorts. We hypothesized falling risk to follow a polygenic mode inheritance, i.e., is influenced by numerous genes with small individual effects¹⁷. Hence, non-GWS SNPs may also contribute to the

genetic component of falling risk. Therefore, PRSs were constructed using PRSice¹⁵ for a series of P-value thresholds ranging from 5×10^{-8} to 1."

We updated the scores range by request from reviewer 1 which now range between 5×10^{-8} to 1

5. The estimates for the genetic correlation of fall susceptibility with fracture and muscle strength mentioned in the text ($r_g = 0.35$ and $r_g = -0.24$, respectively) do not seem to match with what is provided in Figure 3 ($r_g \sim 0.45$ and $r_g \sim 0.15$, respectively). Hence, this should be corrected. Moreover, it would be interesting to know the genetic correlations with alcohol consumption, alcohol dependence and BMI (i.e. these could be included in this Figure), given that the authors used these phenotypes for their Mendelian Randomization analyses.

Reply: We thank the Reviewer for pinpointing these inconsistencies. We have now double-checked the numbers and have corrected them within the text where the lower CI was reported instead of the correlation coefficient. Moreover, we also added in **Figure 3** the genetic correlations for the additional risk factors (BMI, and alcohol dependence).

6. In the Discussion section the authors should mention the lack evidence for eQTL effects for SNPs in the loci on 7p21.3 and 5q21.3, to indicate that it is not yet clear which genes at these loci are implicated in the phenotype (there is currently no mention of this). Moreover, it would be interesting to see the results of a look-up of the lead SNPs at the identified loci in Phenoscanner (<http://www.phenoscanter.medschl.cam.ac.uk/>).

Reply: We want thank the reviewer for these suggestions. We have added the following sentence in the results section: **"However, none of the lead SNPs showed any evidence for eQTL effects."**, and discussion section: **"Overall, none of the lead SNPs show any evidence for significant eQTLs and given the lack of information we cannot claim if the closest genes are also the causal genes."**

Next, we did a look-up of the lead suggestive SNPs ($p < 5 \times 10^{-7}$) and SNPs in LD ($r^2 > 0.8$) in Phenoscanner and GWAS catalogue. A similar search was done for the closest gene and we added the information's to **Supplementary Table 1**. The following was added to the results section:

"Finally, the lead SNP on locus 5q21.2 was previously associated with a variety of traits such as insomnia, depression and neurotism. In addition, several suggestive SNPs showed association with body composition measures such as BMI, fat mass and fat free mass (Supplementary Table 1)."

Reviewer #4 (Remarks to the Author):

It is rather disappointing that you find only two-three loci affecting risk of falling after analyzing more than 90 thousand cases and hundreds of thousands of controls.

Dear Reviewer #4, We thank you for the constructive comments which have helped us improve our manuscript. We were also disappointed with the low yield of discoveries, but now recognize this is part of the genetic characterization of a very heterogenous trait like fall risk.

Please decide if you want to claim two or three loci "genome-wide significant", and if you have replication set or not. According to the text lines 94-96 and Table 2 (see the discovery

part, UK Biobank), there should be two loci. If you want to claim significance as significance after meta-analysis (then you will have at least three loci, as indicated in the abstract, line 197, and Table 1 combined data), I would advise you to do a genome-wide meta-analysis (but then do not use the RS and B-PROOF data set for “replication”).

Reply: We agree with the reviewer that the replication effort was not well defined. To make this clearer we added a separate paragraph about the replication effort on **page 4-5** which reads as follow:

“Replication

We took forward for replication the 17 genome-wide suggestive SNPs ($P < 5.0 \times 10^{-7}$) from the discovery UKBB sample; we sought replication in two smaller prospective population-based studies with older participants, namely the Rotterdam Study (1,009 cases and 4,925 controls) and B-PROOF (1,206 cases and 1,364 controls) cohorts. The B-PROOF Study is a clinical trial on B-vitamin supplements in older adults of advanced age (mean age 74.1 ± 6.5 years) in which fall risk was assessed using retrospective questionnaires at baseline and prospective fall calendars.¹⁵ The Rotterdam Study is a population-based cohort with fall information from participants (mean age 69.5 ± 9.2 years) collected retrospectively from baseline questionnaires.¹⁶ We defined replication as loci harbouring variants associated at a nominal ($P < 0.05$) significant threshold in the replication setting or reaching the GWS ($P < 5 \times 10^{-8}$) threshold in the combined meta-analysis. Overall, the top two SNPs from the discovery phase remained GWS significant in the combined meta-analysis, while replication also brought one additional locus (5q21.3) mapping to RP11-6N13.1 above the GWS threshold (Table 1; Figure 1C).”

1. It is also not quite clear what you have replicated, and what is significance and interpretation of your replication. Quite clearly, you do not see replication of the two loci that were genome-wide significant in discovery. Looking into results of replication of polygenic score (Figure 2) at 5×10^{-7} , I also do not see anything significant. In abstract, you say that “Polygenic risk scores were replicated”; do you refer to the fact that in RS, for PRS derived at p-thresholds 0.01, 0.02 and 0.03 the p-value for association between PRS and the trait is $33 \times 0.001 = 0.033 < 0.05$ (11 thresholds, three tests)? This is not quite clear from the text.

What is interpretation of this finding? That some of the (tens of thousands?) of SNPs that are identified by your GWAS at nominal $p < 0.03$ are indeed related to the fall risk, because a score made of these is associated with $p = 0.001$ in one of the two other studies?

Reply: We apologize for not describing the replication clearly. Please refer to comment 1 above explaining the replication steps. Next, we removed any sentence indicating that the polygenic scores were replicated. We performed PRS analysis in order to estimate the ability of PRSs to discriminate fallers and non-fallers in smaller cohorts representing independent settings. As we hypothesized falling risk to be influenced by numerous genes with small effects (i.e., a polygenic trait), we created scores across different p-value thresholds to test this contention. What we observed is that the best discrimination of fallers and non-fallers in the PRS was achieved when the p-value threshold was relaxed; confirming a polygenic model of inheritance. As such, we added the following text in the Results section (**Page 5**):

“Next, we evaluated the ability of polygenic risk scores (PRSs) constructed from the UK Biobank GWAS results to discriminate between fallers and non-fallers in two independent prospective cohorts. We hypothesized falling risk to follow a polygenic mode inheritance, i.e., is influenced by numerous genes with small individual effects¹⁷ and, hence, non-GWS SNPs may also contribute to the genetic component of falling risk. Therefore, PRSs were constructed using PRSice¹⁸ for a series of P-value thresholds ranging between 5×10^{-8} to 1.”

We updated the scores range by request from reviewer 1 which now range between 5×10^{-8} to 1. That scores constructed using SNP sets from lower thresholds also show association with fall risk in independent settings is also important for assessing the robustness of the genetic correlation analyses, i.e., leverages the association statistic for a much larger number set of SNPs. By demonstrating that these SNPs are in aggregate associated with falls in a separate set of data means that we can put more weight on the genetic correlations.

2. It may be interesting to follow a bit more on significant loci you have identified, e.g. look for pleiotropic effects across other complex trait and “omics” GWAS studies.

Reply: We thank the reviewer for this suggestion. We first did a look-up of the lead suggestive SNPs ($P < 5 \times 10^{-7}$) and SNPs in LD ($r^2 > 0.8$) in Phenoscanner, GWAS catalogue and the GWAS atlas. Similar search was done for the closest gene to the lead SNP and we added the information to **Supplementary Table 1**. The following was added to the results section (**Page 4**):

“Finally, the lead SNP on locus 5q21.2 was previously shown to be associated with a variety of traits such as insomnia, depression and neurotism. In addition, several suggestive SNPs were association with body composition measures such as BMI, fat mass and fat free mass (Supplementary Table 1). “

Minor comments:

Line 73, please fix “variation”

Reply: We corrected this.

In discussion you say (lines 198-199) that “Polygenic risk scores were associated with falling risk in two independent population-based settings”; looking into Figure 2, I see that only results in RS, but not in B-PROOF, pass the multiple testing correction. So, the two “independent population-based settings “mean the discovery cohort of UKB and one of the replication cohorts, the RS? I find this somewhat confusing.

Reply: We have rephrased the sentence to read as *“Polygenic risk scores explained small proportion of the falls variance in two independent population-based settings.”*

Reference:

1. Burgess S, Thompson SG. Interpreting findings from Mendelian randomization using the MR-Egger method. *Eur J Epidemiol.* 2017;32(5):377-389. doi:10.1007/s10654-017-0255-x
2. Bowden J, Del Greco M F, Minelli C, Davey Smith G, Sheehan NA, Thompson JR. Assessing the suitability of summary data for two-sample Mendelian randomization

- analyses using MR-Egger regression: the role of the I2 statistic. *Int J Epidemiol*. 2016;45(6):1961-1974. doi:10.1093/ije/dyw220
3. Bycroft C, Freeman C, Petkova D, et al. The UK Biobank resource with deep phenotyping and genomic data. *Nature*. 2018;562(7726):203-209. doi:10.1038/s41586-018-0579-z
 4. Dudbridge F. Polygenic Epidemiology. *Genet Epidemiol*. 2016;40(4):268-272. doi:10.1002/gepi.21966
 5. Ge T, Chen C-Y, Neale BM, Sabuncu MR, Smoller JW. Phenome-wide heritability analysis of the UK Biobank. Domingue BW, ed. *PLOS Genet*. 2017;13(4):e1006711. doi:10.1371/journal.pgen.1006711
 6. Kemp JP, Morris JA, Medina-Gomez C, et al. Identification of 153 new loci associated with heel bone mineral density and functional involvement of GPC6 in osteoporosis. *Nat Genet*. 2017;49(10):1468-1475. doi:10.1038/ng.3949
 7. Wray NR, Wijmenga C, Sullivan PF, Yang J, Visscher PM. Common Disease Is More Complex Than Implied by the Core Gene Omnigenic Model. *Cell*. 2018;173(7):1573-1580. doi:10.1016/j.cell.2018.05.051
 8. Timpson NJ, Greenwood CMT, Soranzo N, Lawson DJ, Richards JB. Genetic architecture: the shape of the genetic contribution to human traits and disease. *Nat Rev Genet*. 2018;19(2):110-124. doi:10.1038/nrg.2017.101
 9. Purcell SM, Wray NR, Stone JL, et al. Common polygenic variation contributes to risk of schizophrenia and bipolar disorder. *Nature*. 2009;460(7256):748-752. doi:10.1038/nature08185
 10. Wray NR, Lee SH, Mehta D, Vinkhuyzen AAE, Dudbridge F, Middeldorp CM. Research Review: Polygenic methods and their application to psychiatric traits. *J Child Psychol Psychiatry*. 2014;55(10):1068-1087. doi:10.1111/jcpp.12295
 11. Khera A V., Chaffin M, Aragam KG, et al. Genome-wide polygenic scores for common diseases identify individuals with risk equivalent to monogenic mutations. *Nat Genet*. 2018;50(9):1219-1224. doi:10.1038/s41588-018-0183-z

REVIEWERS' COMMENTS:

Reviewer #1 (Remarks to the Author):

I thank the authors for their responses. I only have one further (minor) comment. For the genetic correlation analyses, it is not clear to me how you end up with 46 traits. There are 22 from the medication classes. However, the other 24 traits appear to have been selected a priori (from the hundreds available on LD Hub)? I don't think this is mentioned in the manuscript but is essential information to include.

Reviewer #2 (Remarks to the Author):

Thank you for addressing my concerns in the revised manuscript.

Reviewer #3 (Remarks to the Author):

The authors have addressed all my comments/suggestions satisfactorily.

Reviewer #4 (Remarks to the Author):

I do not agree with the way the authors definition replication. In the era of candidate genes association studies, a relaxed attitude to the questions of significance and replication led to loss of credibility of the field of genetic epidemiology. Most of claims of genetic associations were false, and everyone soon knew that. In contrast, with genome-wide association studies, strict standards were adopted early on, and most of findings - when researchers did follow the standards - turned out to be true. In fact, human GWAS are among sub-fields of life science where the proportion of trustable and replicable findings is high.

[Redacted by editor] For the discovery genome-wide association studies in humans, we claim experiment wise significance at $p < 5e-8$. When N SNPs pass that threshold and we take them forward to replication, we use Bonferroni correction and in the replication studies claim experiment-wise significance if $p < 0.05/N$ is passed. We also ask for direction of effect to be consistent between discovery and replication. One can demonstrate that the rate of false positive claims after following this procedure is rather low, and it does not depend on the number of SNPs taken into replication.

In contrast to the procedure outlined above, the authors define "replication" when suggestive (!) findings pass nominal (!) $p < 0.05$ or a joint analysis $p < 5e-8$. To me, this definition does not follow the high standards that the field should follow if it is to keep its high credibility. The threshold of $p < 5e-7$ corresponds to expedient-wise type 1 error rate of 0.5 [redacted by editor]. If one SNP passes such suggestive threshold (which is an expectation!), and then passes nominal $p < 0.05$, even if we ask for consistency of effects, the rate of false positive claims will be an order of magnitude higher than that following strict definition. Moreover, if more SNPs are taken in such "replication", the false positive rate will grow.

Interestingly enough, in the revised text you do not claim that you have replicated any association, which I agree with ("Overall, the top two SNPs from the discovery phase remained GWS significant in the combined meta-analysis, while replication also brought one additional locus (5q21.3) mapping to RP11-6N13.1 above the GWS threshold"). I also agree that the rs2709062 and rs2111530 are significantly associated. I am less confident that you can really claim that rs2431108 is significant - while it has $p = 4e-8$ in combined analysis, what you do, in fact, is a complicated multiple testing procedure: you test in discovery; select at suggestive threshold; test

in replication; meta-analyse; check p-value in discovery cohort, p-value in replication cohort, and p-value in "meta", and pick up the one which pleases you most... In that, the interpretation of p-value of $4e-8$ (is it experiment-wise significant or not?) is not so straightforward anymore.

We appreciate the comments and suggestions made by the reviewers. Please find our responses to the below.

Reviewer #1 (Remarks to the Author):

I thank the authors for their responses. I only have one further (minor) comment. For the genetic correlation analyses, it is not clear to me how you end up with 46 traits. There are 22 from the medication classes. However, the other 24 traits appear to have been selected a priori (from the hundreds available on LD Hub)? I don't think this is mentioned in the manuscript but is essential information to include.

We agree with the reviewer that including more detailed description of the selection of the additional 24 traits is essential. Indeed, the traits were selected a priori. We have now included the following statement in the methods section (Page 19-20):

“From the variety of traits available on LD-hub we selected 17 cognitive, personality, psychiatric, and neurological traits/disorders a priori that can affect the function of the motor and sensory systems. These traits are essential for the planning and execution of the everyday movements and for the control of posture and balance. Further, we selected seven additional musculoskeletal traits (including fractures; site specific BMD of the femoral neck and lumbar spine and total body; appendicular and total lean mass; and hand grip muscle strength in order to better understand the relationship between falling risk and the risk of fracture. “

“Finally, we also included in the LD-score regression analyses BMI, alcohol consumption and alcohol dependence tested for causal effects on falling risk using the MR approach.”

Reviewer #2 (Remarks to the Author):

Thank you for addressing my concerns in the revised manuscript.

Reviewer #3 (Remarks to the Author):

The authors have addressed all my comments/suggestions satisfactorily.

Reviewer #4 (Remarks to the Author):

I do not agree with the way the authors definition replication. In the era of candidate genes association studies, a relaxed attitude to the questions of significance and replication led to loss of credibility of the field of genetic epidemiology. Most of claims of genetic associations were false, and everyone soon knew that. In contrast, with genome-wide association studies, strict standards were adopted early on, and most of findings - when researchers did follow the standards - turned out to be true. In fact, human GWAS are among sub-fields of life science where the proportion of trustable and replicable findings is high.

[Redacted by editor] For the discovery genome-wide association studies in humans, we claim experiment wise significance at $p < 5e-8$. When N SNPs pass that threshold and we take them forward to replication, we use Bonferroni correction and in the replication studies claim experiment-wise significance if $p < 0.05/N$ is passed. We also ask for direction of effect to be consistent between discovery and replication. One can demonstrate that the rate of false positive claims after following

this procedure is rather low, and it does not depend on the number of SNPs taken into replication.

In contrast to the procedure outlined above, the authors define “replication” when suggestive (!) findings pass nominal (!) $p < 0.05$ or a joint analysis $p < 5e-8$. To me, this definition does not follow the high standards that the field should follow if it is to keep its high credibility. The threshold of $p < 5e-7$ corresponds to expedient-wise type 1 error rate of 0.5 [redacted by editor]. If one SNP passes such suggestive threshold (which is an expectation!), and then passes nominal $p < 0.05$, even if we ask for consistency of effects, the rate of false positive claims will be an order of magnitude higher than that following strict definition. Moreover, if more SNPs are taken in such “replication”, the false positive rate will grow.

Interestingly enough, in the revised text you do not claim that you have replicated any association, which I agree with (“Overall, the top two SNPs from the discovery phase remained GWS significant in the combined meta-analysis, while replication also brought one additional locus (5q21.3) mapping to RP11-6N13.1 above the GWS threshold”). I also agree that the rs2709062 and rs2111530 are significantly associated. I am less confident that you can really claim that rs2431108 is significant - while it has $p = 4e-8$ in combined analysis, what you do, in fact, is a complicated multiple testing procedure: you test in discovery; select at suggestive threshold; test in replication; meta-analyse; check p-value in discovery cohort, p-value in replication cohort, and p-value in “meta”, and pick up the one which pleases you most... In that, the interpretation of p-value of $4e-8$ (is it experiment-wise significant or not?) is not so straightforward anymore.

We follow the contention of the reviewer on this respect. After seeking further statistical advice, we are reassured that the used approach does not invalidate the conclusions of our study, but definitively merits further discussion to address these concerns. While we agree that the associations of rs2709062 and rs2111530 are robust, the reviewer raises concern that rs2431108 is not robust and can represent a false positive association. Therefore, we have added the following text to the discussion on Page 11:

“Next, the combined meta-analysis of all participating cohorts yielded another signal (rs2431108 MAF=0.33) just surpassing the GWS threshold ($P = 4 \times 10^{-8}$). Therefore, the likelihood of a false positive cannot be excluded until additional evidence of replication becomes available, implicating robustly this locus with fall susceptibility.”